# Atrial Fibrillation and Underlying Structural and Electrophysiological Heterogeneity

**DOI:** 10.3390/ijms251810193

**Published:** 2024-09-23

**Authors:** Satoshi Iwamiya, Kensuke Ihara, Giichi Nitta, Tetsuo Sasano

**Affiliations:** Department of Cardiovascular Medicine, Tokyo Medical and Dental University (TMDU), 1-5-45 Yushima, Bunkyo-ku, Tokyo 113-8510, Japan; satoshi-iwamiya.cvm@tmd.ac.jp (S.I.); iharcvm@tmd.ac.jp (K.I.); giichi-nitta.cvm@tmd.ac.jp (G.N.)

**Keywords:** atrial fibrillation, heterogeneity, arrhythmogenicity, electrophysiology, atrial remodeling

## Abstract

As atrial fibrillation (AF) progresses from initial paroxysmal episodes to the persistent phase, maintaining sinus rhythm for an extended period through pharmacotherapy and catheter ablation becomes difficult. A major cause of the deteriorated treatment outcome is the atrial structural and electrophysiological heterogeneity, which AF itself can exacerbate. This heterogeneity exists or manifests in various dimensions, including anatomically segmental structural features, the distribution of histological fibrosis and the autonomic nervous system, sarcolemmal ion channels, and electrophysiological properties. All these types of heterogeneity are closely related to the development of AF. Recognizing the heterogeneity provides a valuable approach to comprehending the underlying mechanisms in the complex excitatory patterns of AF and the determining factors that govern the seemingly chaotic propagation. Furthermore, substrate modification based on heterogeneity is a potential therapeutic strategy. This review aims to consolidate the current knowledge on structural and electrophysiological atrial heterogeneity and its relation to the pathogenesis of AF, drawing insights from clinical studies, animal and cell experiments, molecular basis, and computer-based approaches, to advance our understanding of the pathophysiology and management of AF.

## 1. Introduction

Atrial fibrillation (AF), a common arrhythmia, has been associated with increased risks of heart failure and stroke, consequently exacerbating patient prognoses. Despite its prevalence, AF treatments have not yet been optimized. Pharmacotherapy for AF using antiarrhythmic drugs is characterized by limited efficacy, with no substantial evidence indicating improvements in long-term outcomes [1]. Upstream therapies, such as angiotensin-converting enzyme inhibitors (ACEis) and angiotensin-receptor blockers, have not been proven to prevent the occurrence of AF [2,3]. Despite the effectiveness of catheter ablation for paroxysmal AF, it has therapeutic limitations for persistent AF. As AF progresses from the paroxysmal to the persistent phase, it becomes necessary to focus not only on the triggers but also the vulnerable substrates that drive fibrillatory activity.

The detailed pathophysiological mechanisms underlying AF remain largely elusive. In particular, the processes by which the triggers induce electrical fibrillatory activity and the factors that sustain this abnormal activity remain poorly understood. Furthermore, the determining factors of the seemingly disorganized propagation pattern during AF remain unclear. These mechanisms have been investigated from various perspectives, such as anatomical structure, tissue, sarcolemmal ion currents, and electrophysiology. One of the contributing factors to the vulnerable substrates for AF, namely arrhythmogenic substrate, is heterogeneity. The electrical activity in an atrium is believed to repeat even and synchronous excitement unless heterogeneity exists. When a normal atrial excitation wave advances evenly and synchronously on a homogeneous substrate but encounters a heterogeneous point, it breaks up (wavebreak) and occasionally rotates. If a cascade of wavebreaks is induced, the wave begins to exhibit chaotic propagation. As the propagating wave rotates consistently around the point of the wavebreak, if sustained, it is termed a rotor, which typically has a short cycle length and produces rapid tachycardia. A rotor can be a driver of AF, which represents a focal source demonstrating repetitive and fast activity that propagates outward from the source. The process from the inception of a wavebreak to the sustenance of fibrillation can be attributed to heterogeneity [4]. Many studies have reported a correlation between atrial heterogeneity and susceptibility to AF; however, how these atrial heterogeneities correlate with AF remains unclear, resulting in the unawareness that the underlying heterogeneities are a causative factor of AF. Two main types of heterogeneity exist: structural and electrophysiological. Structural heterogeneity can be further viewed based on three subcategories: macroscopic (segmental), mesoscopic (tissue), and microscopic (cellular and molecular) structure. Each level of structural heterogeneity can affect electrophysiological heterogeneity, contributing to the substrate for AF [5,6,7] (Figure 1).

Highlighting atrial heterogeneity will help us understand its contribution to AF and investigate new therapeutic strategies for AF [8]. Herein, we will summarize the atrial structural and electrophysiological heterogeneities which are incorporated into the pathophysiology of AF, drawing insights from human clinical studies, animal experiments, cell experiments, and theoretical models.

## 2. Macroscopic Heterogeneity

The atria are complex structures composed of various components, including chambers divided by the atrial septum, venous elements, appendages, and vestibules, rather than a single uniform structure (Figure 2). At the boundaries of these segments or within each segment, inherent heterogeneities exist, even in the absence of pathophysiological conditions. In particular, venous components, such as the pulmonary vein (PV), vena cava, and coronary sinus (CS), have common peculiar features. They have in common myocardial tissue stretching from the atrium, which is referred to as a myocardial sleeve. The features of the myocardial sleeve differ from those of the atrial myocardium in terms of myocardial thickness and arrangement. Around the boundary between the veins and atrium, abrupt changes in musculature structure may cause electrical divergence or convergence, possibly resulting in an electrical source–sink mismatch. Meanwhile, the right atrial (RA) appendage is composed of pectinate muscles, which have an uneven and rough surface characterized by the branching and overlapping arrangement of trabeculae. Moreover, the crista terminalis, the connection between pectinate muscles and the relatively smooth muscle region derived from the sinus venosus, generates an electrical boundary. The complex structure of pectinate muscles may contribute to the arrhythmogenic substrates [9]. Although these distinct structural heterogeneities are not arrhythmogenic in most healthy individuals, they become arrhythmogenic and evoke AF when certain pathophysiological conditions arise. The causative factors of AF based on anatomical heterogeneity are listed in Table 1.

### 2.1. Pulmonary Vein

PVs play prominent roles in AF, and catheter ablation for the electrical isolation of PVs has become an established treatment because the majority of AF is initiated by ectopic discharges mostly within the PVs. Abnormal automaticity and triggered activity have been suggested as the triggers of AF. The architecture of the PV myocardial sleeve can be a trigger source for AF, and a few mechanisms have been advocated in canine experiments. Previous anatomic-electrophysiological studies in isolated PV specimens have demonstrated that PVs contain a mixture of pacemakers and myocardial cells [10]. These pacemaker cells with spontaneous activity have a significantly lower density of inward rectifier potassium current, leading to less negative diastolic potential, with which increased automaticity is generally associated [11]. Thus, pacemaker activity from these cells is thought to result in the formation of the ectopic beats that initiate AF. Several studies using the atria of autopsy patients support these findings in canine experiments. Light microscopy with periodic acid–Schiff staining and electron microscopy identified that the PV has cells that are histologically and morphologically similar to node, transitional, and Purkinje cells [12]. Not only enhanced automaticity, but also triggered activity is referred to as impulse initiation in cardiac fibers which is dependent on afterdepolarizations [13]. Animal experiments showed that isolated PV cardiomyocytes (CMs) from healthy dogs and rabbits manifested arrhythmogenic afterdepolarizations, suggesting that triggered activity may account for the trigger of AF within the PV [11,14]. In addition, PVs contain enriched autonomic innervation, and the autonomic nervous system is suggested to enhance ectopic triggers via sympathetic or vagal activation [15,16]. These findings indicate that the PV myocardial sleeve may act as a focal trigger. Furthermore, not only do PVs act as a trigger of AF, but they also contribute to the initiating and sustaining mechanism of AF [17]. The arrhythmogenicity of PVs can be evidenced in humans by persisting tachyarrhythmia confined to the PV during sinus rhythm after its isolation, suggesting that the PV myocardial sleeve is involved in the maintenance mechanism of AF. Furthermore, many of the focal left atrial (LA) tachycardias after segmental PV isolation have been reported to be caused by a focal reentrant circuit located at the PV ostium [18]. From a structural perspective, this is partly attributed to the arrangement and thickness of the myocyte bundles within the PV myocardial sleeves and at the PV-LA junction. The sleeves are composed of circularly or spirally oriented bundles of myocytes with additional bundles that are longitudinally or obliquely oriented [19]. These intricate arrangements of myocardial fibers in the PVs, which are often diagonal or perpendicular to LA CMs near the PV-LA junction, contribute to conduction slowing because of the conduction anisotropy, serving as a possible reason for their complexity within the PV and even at the PV-LA junction [19,20]. According to a histological study performed on human hearts, the thickness of the sleeves is not uniform. The myocardial thickness of the PV is almost uniformly thin (0.3–0.8 mm) in contrast to that of LA (1.1–2.6 mm). An abrupt change in myocardial thickness exists around the PV-LA junction (mean 1.1 mm), and the sleeve tapers distally [21]. The differential muscle narrowing and complex conduction patterns at the PV-LA junction can provide a robust anatomical basis for source–sink mismatch and conduction anisotropy. The arrangement of muscle bundles between different ipsilateral PVs, the carina, is also convoluted and interwoven, which may produce non-uniform anisotropic properties [22,23]. The catheter ablation strategy of PV antrum isolation to encircle ipsilateral PVs by ablating on the atrial side of the PV-LA junction has been widely accepted. PV antrum isolation has achieved a higher clinical success rate and a lower incidence of postprocedural atrial arrhythmias [24], since it can be attributed not only to the isolation of PVs but also to the isolation of the PV-LA junction and the ipsilateral carina. These structural characteristics of the PV myocardium potentially represent a primary substrate for AF.

### 2.2. Vena Cava

Myocardial sleeves within the superior vena cava (SVC) can also serve as the origins of AF [25], and the SVC accounts for the major portion of non-PV foci [26,27,28]. Unlike PVs, limited data regarding the morphology of SVC myocardial sleeves have been published. However, a previous canine study revealed that half of the isolated CMs from SVC myocardial sleeves have pacemaker activity, showing spontaneous depolarization and less negative resting membrane potentials [29]. The presence of CMs with pacemaker activity suggests that automaticity plays a role in the arrhythmogenicity of the SVC. This study also demonstrated that the infusion of autonomic agents, such as isoproterenol, atropine, and phenylephrine, into SVC CMs accelerated the spontaneous activity and induced afterdepolarizations, indicating that the enhanced automaticity and afterdepolarization are involved in the arrhythmogenic activity of the SVC [29]. Additionally, a recent study showed that the muscle fibers of SVC myocardial sleeves exhibit a morphology similar to that of Purkinje fibers [30], supporting the indication of possible arrhythmogenicity in the SVC. In addition to the cell types in SVC myocardial sleeves, the geometric size of SVC myocardial sleeves can be an important element of SVC arrhythmogenicity. In a clinical study, an SVC myocardial sleeve longer than 30 mm has been reported as an independent risk factor for arrhythmogenic trigger sources from the SVC [31]. Long SVC myocardial sleeves were also reported to be correlated with the SVC potential amplitude, reflecting greater myocardial volume. Thus, these findings suggest that long myocardial extension and large amounts of CMs in the SVC can be a source of arrhythmogenicity. SVC myocardial sleeves may serve not only as a trigger of AF, but also as a perpetuator that initiates and sustains AF. Similar to PV myocardial sleeves, the arrangement of musculature in myocardial sleeves within the SVC is intricate. The cardiac fibers are bundled and disposed in longitudinal, oblique, or circumferential directions in the venous wall, which may cause anisotropic conduction within the SVC. Based on high-resolution electroanatomic mapping, slow and heterogeneous conduction has been observed during sinus rhythm and is exaggerated by premature stimulation, indicating that SVC myocardial sleeves have the potential for a reentrant mechanism [32]. Several observations that the SVC acts not only as an AF trigger but also as a driver have been reported [33,34]. Miyazaki et al. reported that SVC fibrillation was confined to the SVC after its electrical isolation, indicating the existence of the driver within the SVC [34]. Therefore, the intricate structure and heterogeneity of the SVC may be involved in the AF substrate, which could potentially facilitate the heterogeneity of the underlying electrophysiological property. For non-responders to PV isolation and patients undergoing repeated ablation, SVC isolation should be considered, as it can separate a highly heterogeneous region with a possible trigger or an AF perpetuator from the RA [35].

The musculature extension into the inferior vena cava (IVC) is shorter than that into the SVC, rendering a lower possibility of an AF source [36,37]. However, a few reports have shown that the IVC serves as a driver for the maintenance of AF in addition to being a trigger [38,39]. The optimal IVC ablation strategy for AF remains unknown because of the limited studies, although focal ablation and IVC isolation have been reported [39].

### 2.3. Coronary Sinus

The CS occupies the atrioventricular groove, and its wall is covered with a myocardial sleeve. The CS is a remnant of the sinus venosus, and its muscle sleeve can be an extension of the RA myocardium over the CS. Histological studies have shown that the CS myocardial sleeve has muscular connections with both RA and LA fibers [40,41]. Muscular fibers of varying thickness arise from the myocardial sleeve along the inferior mitral annulus, providing electrical continuity with the LA. Striated myocardial connections exist between the CS and LA ranging from one to two fascicles surrounded by insulating compartments of adipose tissue [42]. The CS muscle sleeve and its branching connections to the atrium have been implicated in the genesis of atrial tachyarrhythmias. Non-PV ectopic beats arising from the CS have also been reported in clinical studies, suggesting that AF may originate within the CS [27,43]. Little evidence regarding the abnormal automaticity within the CS has been reported; however, some previous reports on CMs isolated from canine CS have confirmed the triggered activity [44,45]. Conversely, several studies have demonstrated that the CS myocardial sleeve may serve not only as a source generating a focal trigger but also as a part of a reentrant circuit [46,47]. The complex orientation of muscular fibers around the CS and their discrete insertion sites within the LA construct a heterogeneous structure and may facilitate the source–sink mismatch, leading to conduction disturbance within the connections [48]. These electrical connections between the CS and both atria may be involved in arrhythmogenesis by forming a part of the reentrant circuit [49]. Optical mapping in a canine model showed a functional conduction block at the RA-CS junction, and this site was involved in the rapid pacing-induced reentrant circuit [50]. Activation maps derived from another canine model demonstrated that the CS musculature developed unstable reentry and AF, which were prevented by the isolation of CS musculature from LA tissue [51]. Similarly, in a clinical study, electrical dissociation of the CS from the LA led to less inducibility of sustained AF after PV isolation, which suggests that the CS is involved in perpetuating AF [46].

### 2.4. Crista Terminalis and Pectinate Muscles

The RA appendage is readily distinguished from the RA because of the crista terminalis and pectinate muscles [52]. The crista terminalis has been demonstrated as a conduction barrier during typical atrial flutter. The investigations of the conduction properties within the crista terminalis demonstrated that it has marked anisotropic conduction with enhanced conduction velocity in the longitudinal direction compared to that in the transverse direction [53]. Meanwhile, in terms of the genesis of AF, animal experiments reported that isolated cells from the crista terminalis in rabbits have spontaneous pacemaker activity, whereas those in dogs showed delayed afterdepolarization induced by norepinephrine [53,54,55]. In a clinical study, the non-PV ectopic triggers of paroxysmal AF originating from the crista terminalis were observed [27]. These findings endorse the view that spontaneous activation may be initiated at the crista terminalis. Moreover, the complex branching structure of the pectinate muscle network has been proposed to provide a substrate for complex patterns of propagation during AF. Pectinate muscles spread throughout the wall of the atrial appendage with a branching and overlapping arrangement. The pectinate muscles are composed of ridges; however, the wall in the groove between the ridges is very thin [36]. This complex structure results in a source–sink mismatch at the branching site of bundles. An electrophysiological study of canine RA tissue showed that a premature stimulus induced conduction delay along the ridge of large pectinate muscles, leading to a wavebreak and resulting in the initiation of reentry [56]. The direction of the propagation in the RA appendage was also demonstrated to be rate-dependent with variability from beat to beat at the site of pectinate bundles, leading to a transformation of fibrillatory conduction [9]. Thus, pectinate muscles may contribute to the vulnerable substrate, allowing the sustenance of AF. Owing to the recent progress in electrode mapping techniques, AF rotational drivers were identified and described within the RA appendage [57,58].

Pectinate muscles are much less extensive in the LA appendage, and the LA appendage lacks a muscular bundle equivalent to the crista terminalis [36]. Despite these anatomical characteristics of the LA appendage being distinct from the RA appendage, the LA appendage has been identified as a possible trigger origin and substrate for AF maintenance after PV isolation [59,60]. Previous studies have demonstrated that LA appendage isolation in addition to PV isolation by catheter ablation reduced AF recurrence compared with PV isolation alone [61,62]. However, procedures with LA appendage isolation had higher rates of thromboembolism compared to those without it [63].

### 2.5. Left Atrial Posterior Wall

The LA posterior wall is a major non-PV trigger that initiates paroxysmal AF [27]. Previous studies have shown that the shortest AF cycle length and highest dominant frequency during AF are found in the LA posterior wall in dogs, sheep, and patients with chronic AF, implying that the LA posterior wall can be a potential source of AF [64,65,66,67]. However, to date, the reason why the LA posterior wall is so arrhythmogenic remains unclear. Of note, the LA posterior wall has a common embryologic origin with PVs, which may contribute to the factors that make the LA posterior wall distinct from other areas of the atrial wall [68,69,70,71,72]. However, the way in which the embryologic origin of the LA posterior wall is associated with the sources of AF remains elusive. On the other hand, an anatomical factor potentially characterizes the arrhythmogenicity in the LA posterior wall, despite the lack of evidence regarding the involvement of abnormal automaticity and triggered activity. The fiber orientation and myocardial thickness of the LA posterior wall abruptly change with partial involvement of the septopulmonary bundle. This bundle arises from the anterior interatrial raphe, ascends obliquely, and combines with longitudinal fibers from the anterior vestibule to run between the left and right PVs on the LA posterior wall, forming heterogeneous fiber orientation [73]. Furthermore, the LA posterior wall exhibits varying degrees of thickness, with the largest increase in myocardial thickness occurring at the border of the septopulmonary bundle [73]. These abrupt changes in fiber orientation and thickness may characterize the electrophysiological properties within the LA posterior wall. Markides et al. demonstrated that a line of functional conduction block in the LA posterior wall exists, running craniocaudally between the PVs during sinus rhythm and PV ectopy in humans [74]. They also demonstrated that patients with AF developed this line of conduction delay, corresponding to a change in fiber orientation in this region. Wavefronts entering the atrium from the PV interact with this functional line of conduction block, resulting in the reentry formation of daughter wavefronts. In an ovine study, electric source–sink mismatch and the subsequent wavebreaks mostly appeared at the septopulmonary bundle near the right superior PV, where the myocardial thickness dramatically expanded [73]. These findings indicate that the structural peculiarity may promote arrhythmogenicity in the LA posterior wall.

To summarize this section, macroscopic views show several common features throughout the atrial region. The pacemaker activity underlying myocardial sleeves within PV and SVC, and the arrhythmogenic afterdepolarization in the myocytes of PV, SVC, and CS, are potential triggers of AF. The intricate and anisotropic arrangement, and varying thickness of muscle fibers can provide an anatomical basis for source–sink mismatch and anisotropic conduction, which are causative mechanisms of the reentrant circuit, located in the PV (including PV-LA junction), SVC, CS-LA junction, crista terminalis, pectinate muscles, and septopulmonary bundle of the LA posterior wall.

**Table 1 ijms-25-10193-t001:** Macroscopic heterogeneities predisposing to AF. RA: right atrium, RAA: right atrial appendage, LA: left atrium, and LAA: left atrial appendage.

Macroscopic Structure					
Region	Structure	Causative Factor	Species and Reference	Mechanism for Arrhythmia	Section
Pulmonary vein (PV)	PV myocardial sleeve	pacemaker activity	dog [10,11], human [12]	arrhythmogenic trigger	2.1
		afterdepolarization	dog, rabbit [11,14]	arrhythmogenic trigger	
		intricate muscular bundle arrangement	human [19]	reentry	
		autonomic innervation	human [15,16]	arrhythmogenic trigger	
	PV-LA junction	varying thickness of muscular fiber	human [19], dog [20]	reentry	
		intricate muscular bundle arrangement	human [21]	reentry	
	carina	intricate muscular bundle arrangement	human [22,23]	reentry	
Superior vena cava (SVC)	SVC myocardial sleeve	pacemaker activity	dog [29], human [30]	arrhythmogenic trigger	2.2
		afterdepolarization	dog [29]	arrhythmogenic trigger	
		intricate muscular bundle arrangement	human [33,34]	reentry	
Coronary sinus (CS)	CS myocardial sleeve	afterdepolarization	dog [44,45]	arrhythmogenic trigger	2.3
	RA-CS connection	functional conduction block	dog [50]	reentry	
	CS-LA connection	intricate muscular bundle arrangement	human [46,47,48]	reentry	
		varying thickness of muscular fiber		reentry	
Crista terminalis (CT)	CT	pacemaker activity	rabbit [54,55]	arrhythmogenic trigger	2.4
		triggered activity	dog [53]	arrhythmogenic trigger	
		anisotropic conduction	dog [53]	reentry	
Pectinate muscle	RAA (LAA)	branching structure (ridge and groove)	human [36], dog [56]	reentry	
Left atrial posterior wall	septopulmonary bundle	intricate muscular bundle arrangement	sheep [73], human [74]	reentry	2.5
		varying thickness of muscular fiber		reentry	

## 3. Mesoscopic Heterogeneity

Anatomical structural heterogeneity at the segmental level, as described above, is inherently present and may become a potential factor of triggers and substrates for AF. However, most AFs are believed to occur when combined with other atrial pathophysiological conditions. Particularly in heart failure, increased atrial volume overload leads to the elevation of atrial pressure and stretching of atrial muscle, resulting in the reconstruction of tissue and the alteration of electrophysiological properties, known as atrial remodeling [6]. Atrial remodeling histopathologically involves the activation and proliferation of myofibroblasts (MFs), which are responsible for the uncontrolled deposition of the extracellular matrix. The enhanced fibrosis leads to tissue anisotropy, and obstructs the electrical wave propagation [75,76]. The pathological sources of AF have been proposed from the view of tissue-level heterogeneity.

### 3.1. Myofibroblasts and Fibrosis

Fibrosis refers to an increased deposition of collagen and other extracellular matrix proteins in the interstitial space. The excessive deposition of collagen enlarges interstitial spaces and decouples electrical cell-to-cell communication at gap junctions, resulting in the reduction in inter-myocyte electrical coupling. As described above, PVs are a major segment that proposes high arrhythmogenicity. An important tissue feature of PVs is the patchy areas of fibrosis, which can be detected even within healthy PVs [19]. In addition, a higher degree of fibrosis in the atrial myocardium extends to the PVs in patients with AF [21]. The inherent and acquired fibrosis within PVs is a possible source of arrhythmogenicity. Furthermore, fibroblast proliferation not only promotes collagen production which serves as an electrical insulator, but also mediates the electrical coupling with CMs. MFs are basically unexcitable cells, but they can form hetero-cellular gap junctions with CMs [77]. Some in vitro experiments using CMs isolated from animal hearts or in silico experiments demonstrated that the electrical coupling with MFs depolarizes the resting membrane potential of CMs [77,78,79]. According to a numerical study, these actions of MFs alter the electrophysiological characteristics of CMs, such as the shortening of myocyte action potential duration (APD) and a decrease in myocyte maximum depolarization velocity, leading to slow conduction velocity [80]. As a result, both the extracellular matrix and MFs bring about further complex electrophysiological changes, potentially causing arrhythmogenicity.

### 3.2. Heterogeneous Distribution of Fibrosis

As atrial remodeling progresses, the heterogeneous distribution of fibrosis is exacerbated, which is one of the factors promoting the pathophysiology of AF. To assess the tissue heterogeneity caused by fibrosis in an entire atrium, one non-invasive method for visualizing atrial fibrosis is cardiac magnetic resonance (CMR) imaging. Late gadolinium-enhanced CMR (LGE-CMR) has been recognized as an indicator to quantify the scar areas in an atrium [81,82], making LGE-CMR a promising tool for visualizing atrial fibrosis. LGE-CMR in patients with AF showed that the fibrotic area is heterogeneously distributed, and preferentially located at the LA posterior wall and around the antrum of the left inferior PV [83,84,85]. Previous studies have demonstrated that more advanced fibrosis detected by LGE-CMR is related to the recurrence of AF after PV isolation [86,87]. Therefore, to investigate the efficacy of catheter ablation targeting atrial fibrosis detected by LGE-CMR, the DECAAF II trial executed the LGE-CMR-guided strategy of identifying fibrotic regions compared with conventional methods for persistent AF [88]. However, no significant difference in the recurrence rate of AF was observed between the two groups. Much debate remains regarding the therapeutic applications based on the LGE-CMR. Another method currently used to estimate fibrotic tissue is electroanatomic voltage mapping. Low-voltage zone (LVZ), defined as an area with less than a certain cutoff using bipolar voltage mapping, has been widely used as a surrogate for local fibrotic tissue [89]. Several previous studies have shown that the most frequent localization of the LVZ tends to be within the LA anterior wall, followed by the septum and LA posterior wall [85,90]. Another observational study suggested that this segmental LVZ distribution pattern advances as AF perpetuates; initially in the LA anterior wall, followed by the LA posterior wall [91]. The severity of fibrosis estimated by the LVZ during sinus rhythm is associated with poor outcomes after PV isolation [92,93]. Several meta-analyses have reported that LVZ-guided ablation for persistent AF provides a significant reduction in recurrence rates [94,95,96]. In a study comparing the fibrotic areas shown by LGE-CMR and LVZ in the LA, LGE-CMR and LVZ covered 55% and 24% of the LA, respectively, while 61% of the LVZ was co-located with the LGE area, and only 28% of the LGE area displayed LVZ [85]. A high mismatch was observed in the distribution and volume of fibrosis detected by the LGE-CMR compared with that detected by the LVZ. The cause of this significant spatial mismatch is still under debate. Considering the association of the fibrotic area detected by these modalities and the electrophysiological properties, both of them correlate with a decrease in atrial conduction velocity [97,98,99]. On the other hand, the relationship between the LGE area and local reentry or rotor remains debated at present. While some reports suggest a higher prevalence of reentry in areas where enhancement is observed on LGE-CMR, other reports do not support this correlation [100,101,102,103]. Similarly, a wide discrepancy exists in the distribution of the LVZ and sites of identified rotors [104]. In fact, several studies have reported that reentrant AF sources were identified at only 8–37% of the LVZ [85,104].

### 3.3. Fibrosis and Arrhythmogenicity

Although the heterogeneous distribution of fibrosis does not necessarily allow for identifying the site of AF sources, the factors causing local conduction disturbances are believed to be partly ascribed to increased fibrosis [105]. Fibrosis has been indicated as the primary factor causing AF. However, among the fibrotic volume, distribution, or both, the factor that contributes the most to arrhythmogenesis remains unclear. Previous studies have implied that the heterogeneity of fibrotic distribution is a more contributive factor to the development of AF [106,107]. In a recent study with a porcine ischemic model, heterogeneous fibrosis in the LA rather than the overall level of fibrosis was associated with an increased AF susceptibility [108]. Several studies have demonstrated that the areas adjacent to dense fibrosis, in relatively healthy zones, show high levels of arrhythmogenic activity [109,110]. Another study also reported that the patchy and heterogeneous distribution of fibrous tissue can lead to conduction delays and the fragmentation of signals [111]. Jadidi et al. reported fractionated continuous activity in the vicinity of the border zone within the patchy fibrotic area, rather than within the dense fibrotic area [109]. However, determining the exact role of fibrotic tissue in arrhythmogenesis experimentally is difficult; therefore, experimental studies have been substituted with some numerical studies. A simulation study demonstrated that an important factor determining the formation and dynamics of arrhythmia in heterogeneous fibrosis stems from the maximum local fibrosis density occurring within the heterogeneous tissue [112]. A highly heterogeneous distribution can induce wavebreaks resulting in the formation of meandering rotors [113]. Other theoretical studies demonstrated that the degree and distribution of fibrosis had a large effect on rotor locations [114], and AF can be perpetuated by rotors meandering in the border zones of patchy fibrosis [115,116]. These numerical findings support the experimental results that the heterogeneous fibrotic distribution provides a certain mechanism of arrhythmogenesis and plays a key role in local fibrillatory activity.

### 3.4. Autonomic Ganglion Plexus

Electrophysiological properties in atria are modified by the autonomic nervous system, in which AF is often mediated by adrenergic or vagal activation [117]. Autonomic ganglia in the atrium are heterogeneously distributed, and ganglion plexuses (GPs) are concentrated at the SVC, CS, PV, and LA posterior wall in humans [118]. Preliminary studies have shown that GP stimulation induces the shortening of APD observed within PV sleeves in dogs [119,120]. Additionally, they demonstrated that injecting acetylcholine into the GP in adipose tissue next to a PV leads to rapid firing from the ipsilateral PV, transitioning to AF, followed by the cessation of AF by ablating the GP at the same site [119,121]. Excessive activity of the autonomic ganglia is now believed to be partially involved in the generation of AF, at least in cases showing focal firing. In clinical practice, attempts have been made to achieve catheter ablation for GP denervation [122]. Recent meta-analyses have shown a reduction in arrhythmia recurrence by adjunctive GP ablation plus PV isolation compared with PV isolation alone [123,124]. When it comes to autonomic nervous activity and AF, the oblique ligament of Marshall (LOM) also plays a key role. It is a remnant of the left SVC and is located between the LA appendage and the left superior/inferior PVs. It runs inferiorly along the inferior atrial wall, while only its intracardiac portion remains patent as the vein of Marshall which drains into the CS. The LOM contains fat and fibrotic tissues, vessels, muscle bundles, nerve fibers, and ganglia. Not only can the LOM’s complex structure serve as an arrhythmogenic substrate, but also its rich innervation by sympathetic nerves may serve as a source of catecholamine-sensitive focal automaticity. Ectopic activity and focal automaticity in the LOM induced by isoproterenol have been demonstrated in dogs, indicating that it contributes to the breakout of AF [125]. The effectiveness of LOM ablation has been reported in patients with persistent AF after PV isolation, implying that enhanced atrial denervation, elimination of AF triggers, or conduction block correlate with a reduction in the vulnerability to AF [126,127].

## 4. Microscopic Heterogeneity

Since cardiac action potentials are governed mainly by sarcolemmal ion currents, the alteration of these ion currents can help understand the arrhythmogenic mechanism. Once arrhythmogenic substrates result in AF, AF further adversely progresses the arrhythmogenic substrates, which is termed “AF begets AF”. In some reports, electrical changes in sarcolemmal ion currents resulting from AF were investigated using a whole-cell patch-clamp system with CMs isolated from the appendages of patients with persistent AF [128,129,130,131,132]. Despite the overall changes in ion currents in AF, to the best of our knowledge, the localized inter- or intra-cellular heterogeneous distribution of ion channels has not been discussed so far. However, a better understanding of the structural and electrophysiological heterogeneity underlying AF at a microscopic level will lead to advancements in treatment strategies. Regardless of the underlying diverse causes, abnormal discharge and electrical reentry are the two main determinants of ectopic firing, initiation, and maintenance of AF. Herein, we will review the microscopic heterogeneities at a cellular level and the molecular basis of the connexins, the profibrotic process, and genetics in association with their predisposition to AF.

### 4.1. Connexin Remodeling and Regulation

As regards the conduction heterogenicity exacerbated by AF, the redistribution of connexin has been mainly argued. Electrical intercellular conduction is highly dependent on connexins, which facilitate cell-to-cell connections (gap junctions) and are known to be involved in the electrochemical coupling to adjacent cells at the intercalated disk [133]. An action potential in a CM propagates longitudinally into the adjacent CM via gap junctions localized normally in the intercalated disk. The heterogeneous distribution in a tissue or lateralization of connexin distribution in a CM has often been discussed. Of note, heterogeneous distribution is termed a patchy distribution in a tissue, and lateralization is termed a lateralized intracellular distribution. Connexin remodeling in a CM, such as reduced expression and lateralization, can decrease longitudinal cell-to-cell electrical coupling resulting in a change in anisotropic conduction. van der Velden et al. investigated the role of gap junctions in a goat AF model [134,135]. They found no changes in the overall gene expression and protein levels of Cx40 and Cx43; however, according to immunohistochemical studies and confocal laser scanning microscopy, a heterogeneous distribution of Cx40 in tissue was observed in AF. Subsequently, Polontchouk et al. demonstrated that in atrial CMs obtained from patients with AF, while the protein level of Cx40 increased, its lateralization in the CMs also increased [136,137]. In patients with AF undergoing a biopsy procedure, the reduced expression and lateralized distribution of Cx40 were observed [138]. Overall, these results showed inconsistent expression levels of Cx40, but the intracellular lateralization of Cx40 was commonly observed. Previous findings on the expression levels and intercellular lateralization of Cx43 in AF were similar to those of Cx40 [133,139]. van der Velden et al. also demonstrated that the total levels and localization of Cx43 remained unchanged in a goat AF model; however, several studies with canine AF models reported increased Cx43 expression together with increased lateralization [140,141]. In humans, a study showed that the LA in patients with lone AF exhibited an increase in the protein expression of Cx43, while another study demonstrated the lateralization of Cx43 in patients with persistent AF [136]. In summary, the heterogeneous expression level and the heterogeneous distribution, including lateralization, of Cx40 and Cx43 may result in heterogeneous intercellular coupling, leading to conduction defects and anisotropy, which can facilitate wavebreaks and pathophysiological substrates of AF [142].

Abnormalities in connexin assembly, permeability, or localization impair electrical propagation and can lead to conduction disturbance which contributes to the development of AF [143]. Those abnormalities manifest in the form of phosphorylation, Cx40/Cx43 protein ratios, and lateralization to the surface membrane [143]. Connexin phosphorylation has been thought to switch on several types of functions. For instance, it regulates gap junctional protein trafficking and assembly. The increase in lateralized connexins can lead to a reduction in gap junctional conductance. Cx43 is phosphorylated specifically by various kinases including protein kinase A (PKA), protein kinase C (PKC), Ca^2+^/calmodulin-dependent kinase II (CaMKII), and mitogen-activated protein kinases (MAPKs) [144]. The sympathetic and β-adrenergic induction of activated cyclic adenosine monophosphate/PKA promote the synthesis and the phosphorylation of Cx43 (Ser364) [145]. PKA enhances the assembly of connexins into the gap junction and suppresses proteolytic degradation. As a result, PKA increases channel conductance. On the other hand, angiotensin II (Ang II) induces PKC activation, which augments the phosphorylation of Cx43 (Ser368) [146], inhibits the assembly of the connexins into the gap junction, and accelerates proteolytic degradation, followed by the impairment of the gap junctional communication. A remarkable reduction in connexins in the intercalated disk, together with an increase in the lateralized delocalization, attenuates cell-to-cell coupling. Ca^2+^ is also an important factor that decreases the gap junctional conductance. Ca^2+^ overload elevated the resistance of gap junctions by hindering the PKA-mediated phosphorylation of Cx43. In addition, Ca^2+^/calmodulin binding activates CaMKII. Not only does CaMKII phosphorylate the ryanodine receptor 2 and increase its Ca^2+^ sensitivity to exhibit Ca^2+^ leak and triggered activity via an increase in channel open probability, but it also exerts an indirect effect by regulating Cx43 expression and subcellular localization in the intercalated disk [147]. Furthermore, inflammatory cytokines activate MAPK to upregulate phosphorylated Cx43 and impair cell-to-cell communication because Cx43 becomes extensively dispersed at the intercalated disks [148,149]. Taken together, the phosphorylation of connexins causes the abnormal distribution and expression of the gap junction, which promotes conduction heterogeneity. Remodeling of the gap junction can be an arrhythmogenic substrate.

### 4.2. Molecular Mechanism of Atrial Fibrosis

Atrial fibrosis is a major factor contributing to atrial structural and electrophysiological heterogeneity. The proliferation of the extracellular matrix (ECM) is promoted by the neurohormonal dysregulation of the renin–angiotensin–aldosterone system (RAAS) and the activation of fibrotic pathways, mainly initiated by the transforming growth factor beta (TGFβ), connective tissue growth factor, and platelet-derived growth factor [150]. These profibrotic signaling molecules activate fibroblasts, resulting in proliferation and differentiation into secretory myofibroblasts, often accompanied by the upregulation of matrix metalloproteinases (MMPs) and the downregulation of the tissue inhibitors of metalloproteinases (TIMPs). AF can activate inflammatory cells, such as macrophages, leading to oxidative stress and the production of reactive oxygen species (ROS). Consequently, the RAAS is activated, and Ang II contributes to a profibrotic process by binding to Ang II type 1 receptor [151], followed by the further stimulation of phospholipase C, which acts through inositol 1,4,5-trisphosphate (IP3) and diacylglycerol (DAG). IP3 mediates an increase in Ca^2+^ levels in the cytoplasm. The intracellular Ca^2+^ overload promotes fibroblast proliferation and differentiation. DAG activates PKC, which activates the downstream signaling of MAPK. Ang II serves as a potent nicotinamide adenine dinucleotide phosphate oxidase activator, leading to ROS overproduction, which activates MAPK as well. The activation of the MAPK signaling pathway promotes the secretion of various transcription factors such as TGFβ and MMP. TGFβ binds to serine/threonine kinase receptors, accompanied by the suppressor of mother against decapentaplegic protein-mediated signal transduction, thereby further promoting the profibrotic process. These processes in cooperation accelerate the formation of vulnerable substrates for AF. In terms of the association between the profibrotic molecular process and heterogeneity, AF vulnerability has been demonstrated to be associated with a segmental heterogeneous pattern of atrial ECM remodeling (molecular MMP/TIMP profile) in a porcine AF model with myocardial infarction [152]. This result implied that a substantial regional heterogeneity exists at the molecular level. Furthermore, atrial inflammation and adipose tissue depots play an important role in AF substrate formation [153,154]. In particular, many studies have suggested an association between epicardial adipose tissue (EAT) and the fibrosis of the underlying atrial myocardium [155,156,157]. EAT contains various cell types, especially adipocytes and immune cells such as macrophages and lymphocytes. Both the immune cells and adipocytes within EAT can release various cytokines, contributing to pathogenic inflammation. In patients with coronary artery disease, EAT has increased levels of inflammatory cytokines [158]. Another study with resected LA appendage and associated EAT from cardiac surgery patients with AF demonstrated that the fibrosis of EAT was associated with LA fibrosis [157]. The fibrosis and inflammation of EAT can predispose to the fibrosis of the adjacent atrial myocardium through the proinflammatory and profibrotic bioactive secretome from EAT [154]. Adipocytes can also have a remote effect on the myocardium by the secretion of adipokines. Profibrotic secretome analysis identified an adipokine, activin A, which is a member of the TGFβ superfamily. Ventedlef et al. demonstrated that atrial fibrosis was promoted with a high concentration of activin A secreted from adipocytes [155]. Taken together, a great amount of EAT correlates with the development of AF through the profibrotic secretome. In terms of the distribution of EAT in the LA, a study evaluating the location of EAT using computed tomography in patients with AF showed that large clusters of EAT are observed adjacent to the anterior roof, LA appendage, and lateral mitral isthmus [159]. This result implies that the heterogeneous distribution of EAT can be a causative factor of the regionally dependent severity of fibrosis in atria.

### 4.3. Genetics of Atrial Fibrillation

Many studies have demonstrated that AF has substantial genetic factors predisposing to the development of AF [160]. The linkage analyses of AF have shown various causative genes for familial AF. Mutations in the potassium channels were discovered early as causes of familial AF: *KCNQ1*, which encodes a subunit of KvLQT1; *KCNH2* which encodes HERG; and *KCNJ2* which encodes Kir2.1 [161,162,163]. An increase in potassium currents due to a gain of function, which shortens the action potential and refractory period of atrial CMs, forms a reentry substrate that promotes the occurrence and maintenance of AF. However, AF is rarely a monogenic disorder; AF is often caused by environmental factors and multiple common genetic predispositions that affect AF susceptibility. Genome-wide association studies have clarified the genetic polymorphisms strongly associated with AF. Multiple variants were found in genes responsible for cardiac development (*NKX2-5*, *TBX5*, and *PITX2*) and electrophysiology, including the coding genes of potassium ion channels (*KCNH2* and *KCNJ2*) and connexins (*GJA1* and *GJA5*) [164]. Although the susceptibility to AF is different across ancestries, some of the variants found across different ancestries are concentrated in the 4q25 region of the long arm of chromosome 4, in which the transcription factor *PITX2* is closest [165]. *PITX2* is expressed exclusively in the LA and has a critical function of regulating the left–right differentiation of the embryonic heart [166,167]. The asymmetrical organ morphogenesis by *PITX2* confines the sinoatrial node, pacemaker cells, to the RA. *PITX2C* is the major isoform expressed in the heart, particularly in the LA, and *PITX2C* expression was significantly downregulated in human patients with AF [168]. Thus, it is probable that a deficiency in *PITX2* causes the incomplete suppression of pacemaker activity in the LA, resulting in enhanced pacemaker activity in the LA. Furthermore, a transgenic murine model with a *Pitx2c* deficiency, which exhibited a decrease in *Pitx2c* expression in only the LA compared to the wild-type, showed the shortening of the LA action potential, and more susceptibility to AF compared to the wild-type [169]. Similarly, the chamber-specific *Pitx2* conditional mutant mice revealed that the LA displayed a more depolarized resting membrane potential and a smaller action potential amplitude [168]. *PITX2* has also been reported to regulate the extension of myocardial sleeves into PVs from the LA during development [170]. The differentiation of dorsal mesenchymal cells into the pulmonary myocardium requires *PITX2*, and transgenic *Pitx2*-deficient mice were found to have normally developed PVs, but absent PV myocardial sleeves [170]. In summary, these findings indicate that genetic predisposition increases susceptibility to AF, either electrophysiologically or anatomically. The regionally heterogeneous expression patterns of these genes can facilitate the basis of electrophysiological heterogeneity.

## 5. Electrophysiological Heterogeneity

The progression of AF is believed to require an ectopic or reentrant basis. When a focal discharge exceeds the threshold to excite the surrounding muscle, the propagation from this ectopic focus possibly acts as a trigger. If the electrophysiological heterogeneity meets the sufficient need for the AF substrate, the wavefront of the propagation wave can trigger a cascade of wavebreaks, leading to continuous reentry and the initiation of AF. The maintenance of continuous activity depends on the existing electrophysiological properties with refractory and excitability determinants; shortened refractory period and reduced conduction velocity (CV). As described above, segmentally differentiated tissue properties, fiber arrangements, and redistributed intra- and inter-cellular connexins facilitate the structural heterogeneities, which in turn connect with electrophysiological heterogeneities in the refractoriness and conduction of action potential.

### 5.1. Repolarization Heterogeneity

The heterogeneity of repolarization in the atria has been reported as a source of AF [171], similar to that in the ventricles where repolarization heterogeneity has been widely reported to cause ventricular tachycardia and fibrillation [172,173]. The analyses of atrial action potentials derived from the patch-clamp technique, monophasic action potential (MAP) method [174], endo- or epicardial electrophysiological studies, or optical mapping [175] are helpful to assess atrial repolarization. According to studies that used these methods, the heterogeneity in the APD or effective refractory period (ERP) serves as a substrate for AF [176,177,178]. In a study comparing action potential morphology using isolated CMs from various atrial segments in adult or aged dogs, and with sinus rhythm or chronic AF, patch-clamp recordings showed a tendency for APD shortening and increased APD heterogeneity in aged dogs with chronic AF compared to that in adult dogs with sinus rhythm [179]. Another canine study investigating the specific cellular electrophysiological properties using a patch-clamp system showed that resting membrane potential was more depolarized, and the APD was shorter in PVs than in the LA [180]. MAP studies in patients with paroxysmal AF showed a significant increase in the segmental dispersion of APD and ERP compared to those in patients with sinus rhythm [171,181,182]. In a canine model, an electrophysiological study demonstrated that the regional heterogeneity of refractory periods increases in AF [176]. Optical mapping with isolated hearts of diabetic rats demonstrated that an increase in the spatial dispersion of APD corresponds to the vulnerability of atrial arrhythmia [183]. Another optical mapping study in canine models showed the progressive shortening of APD and the depolarization of the resting membrane from the LA to the distal PVs [120]. This study also substantiated that the heterogeneity of the APD in PVs provided a favorable substrate for reentry formation. Simulation studies have supported these findings that the heterogeneity in refractoriness can account for the initiation of AF [176,184]. They revealed the presence of segmental dispersion or heterogeneity of APD in chronic AF and even paroxysmal AF. Further studies numerically showed that wavebreaks occur at sites with the steepest gradients of APD dispersion [185,186,187]. Another computer-based study with a 3D human model, which incorporated the regional distribution of APD from previous experimental data, showed that marked regional differences in the APD co-exist with shortened APD, facilitating the initiation and maintenance of reentrant waves. In particular, reentrant excitation is stabilized in the presence of pronounced regional differences in APD at the PV-LA junction and the crista terminalis/pectinate muscle junction in the RA appendage [188].

Many reports have suggested that an increase in inward rectifier potassium current is crucial in AF by shortening repolarization and causing hyperpolarization [128,129,130,131,132]. Consistent with the dispersion of APD, an in vitro assay with isolated CMs from different segments in an atrium showed the heterogeneous expression of the inward rectifier potassium channel [189,190]. Since the functional effects of structural remodeling induced by AF on the regional heterogeneity of APD have not yet been elucidated, computational models have been proposed to characterize these relationships. Several theoretical studies have compared the remodeled ionic currents with AF-induced reduction in APD in a human atrium, demonstrating that the up-regulation of inward rectifier potassium current has a greater influence on APD than any changes in other ion currents [191]. Moreover, Berenfeld et al. introduced the ionically regional gradient model to investigate the relation between the distribution of various types of ion channels and rotor stability. Among them, the gradient of the inward rectifier potassium current had the most significant impact on rotor stability [192,193]. The variation in the conductance of the inward rectifier potassium current resulted in gradients of APD, leading to the trapping of rotors at sites with the steepest gradients of APD [114].

Of note, what complicates the understanding of the reentrant mechanism is that the determinants of electrophysiological properties may change, not only spatially but also temporally [194,195,196]. It is not easy to thoroughly visualize spatio-temporally heterogeneous electrophysiological factors. APD is not just a static electrophysiological parameter but a dynamically changing one that can be influenced by the preceding diastolic interval (DI). If the preceding DI is short, the following APD tends to shorten, whereas if it is long, the following APD tends to be prolonged. The relationship between APD and DI is defined as APD restitution [197]. A steep curve in the APD restitution slope can cause oscillations in APD alternans, a phenomenon in which APD (or repolarization time) alternates between longer and shorter durations over time. When APD alternans occurs in a spatially inconsistent pattern, it is termed discordant APD alternans. Discordant alternans is a phenomenon in which two spatially distinct regions exhibit APD alternance of opposite phases. At the boundaries between the regions in which APD oscillates discordantly, there turns out to be an increased APD gap. The APD gap can induce the functional conduction block at the border, and thus yield heterogeneous conduction patterns, resulting in a possible source of fibrillatory conduction. Previous studies have demonstrated that discordant alternance is associated with reentrant activation and susceptibility to AF [198,199,200]. This substantiates that APD possesses the property of spatio-temporal variability, which can be a cause of electrophysiological heterogeneity and serve as the substrate for arrhythmogenicity. Taken together, repolarization heterogeneity can be a strong determinant of arrhythmogenic substrate.

### 5.2. Conduction Heterogeneity

Atrial conduction disorder can be decomposed into two elements: altered conduction propagation (direction) and reduced magnitude (velocity). When longitudinal propagation along the muscular fiber is disturbed because of the reduced electrical coupling between myocytes or collagenous obstacles, conduction in the transverse direction through lateral cell connections gets involved in the propagation. Additionally, tissue discontinuities due to fibrosis or anisotropic fiber orientation can cause electrical source–sink mismatch along these pathways, resulting in the local slowing of conduction or conduction block, and the tortuous conduction pathway. Furthermore, the hetero-cellular electrical coupling of myofibroblasts and myocytes can increase heterogeneity in excitability, refractoriness, and electrical load, potentially inducing conduction slowing [201]. This can lead to a pivoting or zigzag trajectory of activation. Electroanatomic maps during sinus rhythm of patients with paroxysmal or persistent AF showed that the atrial conduction properties in persistent AF were characterized by a higher wavefront curvature and a larger number of pivoting points than those in paroxysmal AF [202]. Several animal studies with optical mapping recognized zigzag conduction in pathological states such as stretch-induced and age-related atria [203,204]. Slow conduction with pivoting or zigzag pathways has become a factor of localized reentrant circuits [204]. Slow conduction or pivoting points can be viewed as fragmented potential or complex fractionated atrial electrograms (CFAEs) in the local potentials obtained from the electrode catheter during catheter ablation procedures [205,206]. Ablating the localized regions that represent CFAE in addition to PV isolation has been advocated to increase the procedural success rate. A meta-analysis demonstrated that the adjunctive CFAE ablation could provide additional benefits, such as reducing the recurrence of AF for patients with persistent AF [207]. Of note, a CFAE during AF is not always an indicator of an abnormal substrate; however, CFAE during sinus rhythm is a preferential marker of slow conduction, wavefront collision, or pivoting sites. Furthermore, to understand atrial arrhythmogenicity, transmural conduction should also be considered. de Groot et al. demonstrated the asynchronous activation of the endo-epicardial wall during AF in humans using simultaneous endo-epicardial mapping, which was not apparently observed during sinus rhythm [208,209]. These results implied that the dissociation of endo-epicardial activation may be important in the maintenance of AF. Optical mapping studies are helpful to assess the localized CV and visualize the heterogeneity. In animal experiments, atrial heterogeneous CV has been reported using isolated hearts in models such as pressure-overload mice [210], diabetic rats [183], and rabbits with atrial stretch stress [211]. These studies substantiated that incremental atrial heterogeneity in CV is closely related to the increased occurrence of AF. In addition, just as APD depends on the preceding DI, CV depends on it as well [212]. This dynamic is referred to as CV restitution, similar to APD restitution. When static conduction abnormalities are combined with dynamic conduction abnormalities due to CV restitution, conduction becomes more heterogeneous. In other words, the source of AF is present in regions where the decrease in atrial CV and the increase in the heterogeneity of CV with rate dependence are prominent [210,213,214] (Figure 3). To date, while a positive relation between the severity of the conduction heterogeneity and the susceptibility of AF has been suggested, how the conduction heterogeneity is linked to the initiation and sustenance of AF and the propagation pattern during AF remains unclear.

### 5.3. Ectopic Excitement

Ectopic premature contractions are widely acknowledged to serve as critical triggers for AF. The underlying mechanism by which these focal discharges act as triggers is their ability to induce conduction disturbances, thereby manifesting the AF substrate. Mapping studies have demonstrated that an atrial extrasystole can cause a functional conduction block [215,216]. Not only can a premature firing with a short-coupling interval trigger the initiation of AF via heterogenous APD or CV restitution as described in the previous sections, but conduction-directional effects can also predispose to functional conduction block and conduction slowing. In terms of the directional dependence of conduction, Kumagai et al. showed that the conduction delay from the distal PV to the PV-LA junction was significantly longer than that from the PV-LA junction to the distal PV, and a short-coupled extra-stimulus from the PV formed a PV-LA reciprocating reentrant circuit involving exit and entrance breakthrough points at the PV-LA junction [217]. Spach et al. demonstrated with human and canine models that premature stimuli resulted in slower conduction in the transverse direction, leading to anisotropic conduction [218,219,220]. They showed that ectopic premature excitation provokes more conduction disorders and reentries. An intra-operative epicardial mapping during 503 premature atrial contractions revealed that the CV decreases within the PV area, and the local directional conduction heterogeneity increases [216]. Furthermore, they demonstrated that patients with AF have a slower CV and more conduction heterogeneity during sinus rhythm, which becomes more pronounced during premature atrial contractions than in patients without AF.

## 6. Clinical Applications for Reducing Heterogeneity

Three therapeutic approaches can be considered to alleviate the heterogeneity. The first approach is to electrically separate the area with severe heterogeneity from the healthy area (Figure 4A). Isolating not only the trigger source but also regions with significant heterogeneity that deeply contribute to the initiation and maintenance of AF can be considered beneficial. In this context, PV isolation, PV antrum isolation, SVC isolation, and LA posterior wall isolation have been performed, with demonstrated efficacy. To further improve this approach, developing novel procedural techniques and devices for isolating anatomically challenging regions and advancing imaging and mapping technologies for more accurate identification of areas that need isolation is essential. While isolating a larger area can potentially be more effective by surrounding a greater amount of AF substrate with a conduction block, the isolated atrial regions lose their physiological contractile function, which can lead to thrombus formation. Therefore, the isolation should be limited to a necessary and sufficient extent. The second way is to ablate the individual distinctive features contributing to triggers, drivers, or other factors that are considered strongly correlated to AF sources using catheter interventions (Figure 4B). To this end, catheter ablation targeting sites with distinctive local features, such as reentrant circuit with conduction delay [221,222], fragmented potential [206,207], ganglionated plexi [122], local driver [206], and rotor [223], have been developed. However, the indicators to be targeted to achieve optimal electrical modification remain unclear. Particularly in the cases of persistent AF, where target regions are widely distributed throughout the atria, the complete ablation of all the target areas is difficult. Furthermore, the ablation process itself can contribute to myocardial damage and the formation of new conduction disturbances, presenting numerous challenges that need to be addressed. In this approach, developing new methods to identify the required and sufficient targeted areas and innovative interventions to modify myocardial conduction properties without proposing another source of AF is highly desired. The third approach is to decrease the heterogeneity or restore it to the original state with pharmacotherapy (Figure 4C). The development of new pharmacotherapy to reduce heterogeneity, including the profibrotic process or connexin remodeling, which is acquired under pathological conditions, is anticipated. To date, several drugs for atrial arrhythmia and heart failure have been proven to reduce the atrial heterogeneity in animal experiments: sodium channel blockers [217], calcium channel blockers [224], ACEis [225], and neprilysin and angiotensin receptor blockers [210]. However, these drugs have not been proven to reduce the occurrence of AF in clinical settings. While a new treatment is being strongly anticipated, sodium–glucose cotransporter-2 (SGLT-2) inhibitors are currently used for heart failure, and their effectiveness on AF has been gradually uncovered [226]. In short, therapeutic interventions for advanced AF are not established yet since the treatment outcomes decline as heterogeneity is exacerbated. Thus, finding strategies through catheter-based procedures or pharmacotherapy to reduce the heterogeneity could potentially lead to improved treatment outcomes. In addition to elucidating the underlying cause, a new approach to heterogeneity is anticipated.

## 7. Conclusions

The experimental and clinical evidence that the structural and electrophysiological heterogeneity contributes to the substrate for AF is accumulating. Advances in the recognition of the underlying heterogeneity with mapping technologies have elucidated the substrates incorporated into the pathophysiology of AF. Further studies on the underlying heterogeneity would provide more profound insights into the mechanisms of the sustenance and dynamics of AF.

## Figures and Tables

**Figure 1 ijms-25-10193-f001:**
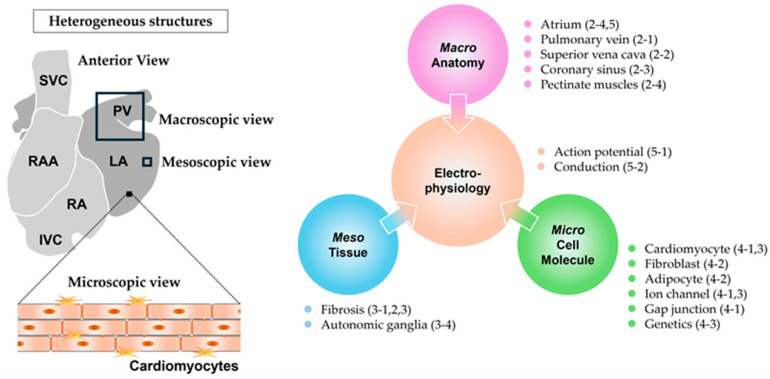
Macro-, meso-, microscopic structural and electrophysiological heterogeneities. RA: right atrium, RAA: right atrial appendage, LA: left atrium, PV: pulmonary vein, SVC: superior vena cava, IVC: inferior vena cava, Macro: macroscopic, Meso: mesoscopic, and Micro: microscopic. Numbers in parentheses indicate the corresponding section numbers in the text.

**Figure 2 ijms-25-10193-f002:**
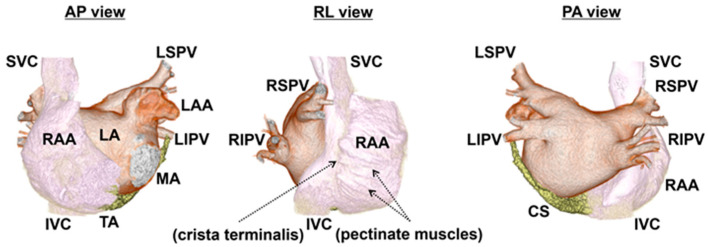
Anatomy of human atria in 3D computed tomography. The endocardial aspects of the atria and surrounding structures above the mitral and tricuspid annulus are cropped using 3D computed tomography. With the 3D images of contrast-enhanced computed tomography reflecting the endocardial aspects of the atria, the endocardial structures, crista terminalis, and pectinate muscles, in particular, are recognizable. Of note, the endocardial aspects of an RAA demonstrate a rough surface due to pectinate muscles, while the body of an RA has a smooth surface. The marks of the crista terminalis and pectinate muscles from the endocardial aspect are shown by the arrows. The images are separated by colors: RA, SVC, and IVC (white); LA, LAA, and PVs (brown); and CS (yellow). AP view: anterior–posterior view, RL view: right lateral view, PA view: posterior–anterior view, RA: right atrium, RAA: right atrial appendage, LA: left atrium, LAA: left atrial appendage, LSPV: left superior pulmonary vein, LIPV: left inferior pulmonary vein, RSPV: right superior pulmonary vein, RIPV: right inferior pulmonary vein, SVC: superior vena cava, IVC: inferior vena cava, CS: coronary sinus, MA: mitral annulus, and TA: tricuspid annulus.

**Figure 3 ijms-25-10193-f003:**
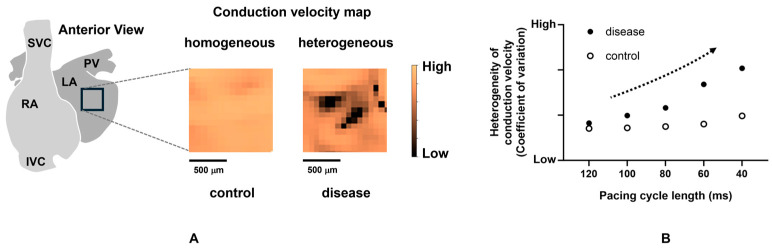
Increased heterogeneity with rate dependence in a diseased atrium. (**A**) Conduction velocity maps (during sinus rhythm) in a control and a diseased murine atrium with heart failure. (**B**) Increase in heterogeneity of conduction velocity dependent on pacing cycle length in a diseased atrium. The dotted arrow indicates an increase in the heterogeneity of conduction velocity. Taken from the reference [210] with partial modification.

**Figure 4 ijms-25-10193-f004:**
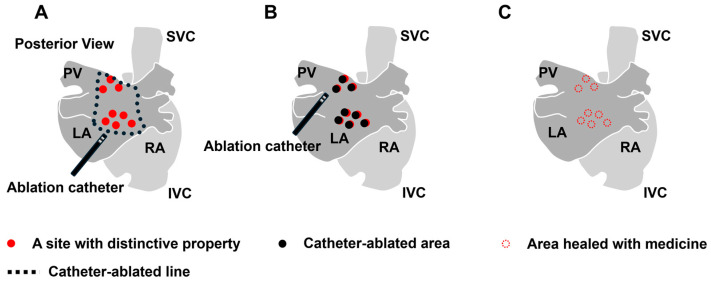
Therapeutic approach of ablation or pharmacotherapy for the heterogeneity. (**A**) Isolation strategy by ablation. (**B**) Ablation strategy for the individual structural and electrophysiological features. (**C**) Pharmacotherapy to reduce the overall heterogeneity by inhibiting the progression of pathophysiology such as profibrotic process or connexin remodeling. RA: right atrium, LA: left atrium, PV: pulmonary vein, SVC: superior vena cava, and IVC: inferior vena cava.

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
