# Peer review of "Atrial Fibrillation and Underlying Structural and Electrophysiological Heterogeneity"

_ijms, 2024, doi:10.3390/ijms251810193_

Round 1
Reviewer 1 Report
Comments and Suggestions for Authors
N/A. I attached the peer review PDF file.

Author Response
Major points
Comments(1-1)
L54-56: The process from the inception of a wave break to the sustenance of the fibrillation can be attributed to heterogeneity [4]. Many studies have referred to the correlation between atrial heterogeneity and susceptibility to AF. Please provide and emphasize the novelty and advantage of the authors’ manuscript reviewing heterogenic mechanisms of AF, compared to the previous literature. For readers of this general science journal, it is readily a little difficult to understand whether focusing on heterogeneity and/or the addition of updated recent findings are novel, unique or worth being summarized as a peer-reviewed article.
Response
We appreciate your kind suggestions. We provided a comment to emphasize the novelty and advantage of reviewing the heterogeneity. (Page 2, Line 53-56)
Comments(1-2)
In the nearly last part of the manuscript, molecular mechanisms and heterogeneity are introduced, such as ion channels, gap junction channels, intracellular Ca2+ and ionic signaling and homeostasis. Is this included in “Microscopic view”? For now, “Microscopic view” looks like a conceptual layer of cellular and its surrounding environment. If it is clearly included, it would be comprehensive for the entire manuscript.
Response
We appreciate your helpful comments. We reconstructed the classification of structural heterogeneity to be comprehensive for the entire manuscript. Therefore, we showed that the microscopic views include cellular and underlying molecular aspect in order not to be confusing for readers (Page 2, Line 58-59), modified the Figure 1 (Microscopic structure) and integrated the section 6. “Molecular basis predisposing to atrial fibrillation” into the section 4. “Microscopic heterogeneity”. (Comment 4-1)
Comments(1-3)
Could the authors put the section numbers next to each subject in Figure 1 shown here to correspond this conceptual depiction to the contents of the review? It could be helpful for readers of this journal.
Response
Thank you for your advisory comments. We changed the figure in that numbers in parentheses indicate the corresponding section numbers in the text. (Page 2, Figure 1)
Comments(2-1)
In this section, structural heterogeneity and AF are very detailed and approximately 5 pages out of 19 pages (>25%, not including references) are used. It would be helpful if there were figures and/or tables explaining this section. If additional figures are allowed, the figure could help understand better spatial anatomical atrial heterogeneity and how AF can spatially occur in atria. This section introduces in detail about structural heterogeneity in various atrial regions and mentions species. The table summarizing atrial structure linked to, for example, AF, its mechanism(s) (idiopathic, disease- or surgery-induced, etc.), used species, reference numbers, etc., could be very helpful for readers to understand atrial structural heterogeneities and AF. For example, this review effectively uses figures and tables as below. Atrial fibrillation. Nat Rev Dis Primers. 2: 16016. doi: 10.1038/nrdp.2016.16. PMID: 27159789. Tables concisely summarize multiple categories in good order. Figures provide spatial and functional information synergistically. The main part is approximately 21 pages and 10 figures and 5 tables are approximately 8.5 pages out of it.
Response
Thank you for your constructive comments. We inserted Table 1 to summarize the section 2. (Page 4)
Comments(2-2)
This section introduces and ends with specifics of atrial structures and the pathophysiology of AF. Is it possible to summarize these individual points at the end of the section?
Response
Thank you for your advice. Surely yes, it is. We summarized the points of macroscopic views at the end of this section 2-5. (Page 9, Line 310-317)
Comments(4-1)
The space taken in this section for one of the three components of atrial heterogeneity introduced in Figure 1 is too short compared to “Macroscopic view” using 5 pages and “Mesoscopic view” using 3 pages. This section mainly introduces gap junction channels, although according to the right part of Figure 1, “Microscopic view” includes ion channels, gap junctions, etc. The ion channels are discussed in section 6. It is currently confusing for readers according to the concepts given in Figure 1.
Response
We appreciate your constructive comments. We integrated the section 6. “Molecular basis predisposing to atrial fibrillation” into the section 4. “Microscopic heterogeneity” to make this article more consistent according to the concepts in Figure 1. (Page 14, Line 508- Page 16, Line 626)
Comments(4-2)
According to Figure 1, “Microscopic view” indicates cardiomyocytes, fibroblasts, etc. If noncardiomyocytes can be included, it might be possible to include immune cells, adipocytes, etc. Another question is: are the following not the focuses of heterogeneity in this section/manuscript? Disrupted or heterogeneously localized membrane proteins and intracellular structures in affected atria.
Response
Thank you for your suggestions. We inserted the issues of adipocytes and immune cells in association with profibrotic mechanism. (Page 15, Line 567-583). Disrupted or heterogeneously localized connexin are the focusses of heterogeneity as mentioned in the section 4-1. We emphasized the heterogeneous expression and localization of connexin according to the issue for more clarity. (Page 14, Line 503-507)
Comments(4-3)
Based on (4-1) and (4-2), the volume of contents of each heterogeneity layer and the correspondence of the contents and concepts shown in Figure 1 would be arranged.
Response
We appreciate your constructive advice. We revised the article according to your comments 4-1 and 4-2.
Comments(5-1)
L580. The end of the paragraph in section 5-2 seems missing, at least in the given PDF manuscript file. Please fix it.
Response
Thank you for your comments. We fixed it.
Comments(5-2)
Figure 2, right graph. Is this a conceptual figure or one based on the actual values of humans or the specific species? Although the descriptions of high-low and long-short are very conceptual, this graph contains scales in between, suggesting that there are some values indicated by them. If this is based on generalized values, the x and y axes could be given with actual values unless multiple species are mixed. Dots, instead of fit lines, on the scales also lead readers to think that there are values from the real-world data.
Response
Thank you for your accurate comments. Figure 3 (right panel) shows actual figure based on our research (taken from ref. [210]). We revised the figure by showing actual values on x-axis. In terms of y-axis, showing the actual values of coefficients of variation seems difficult to understand, and thus high/low heterogeneity remains shown for more clarity. (Page 20)
Comments(6-1)
In this molecular section, the appearance of the words “heterogeneity” and “heterogenous” is much lower than in other sections where they are used to explain AF. How could molecular mechanisms initiate and lead to AF with this manuscript’s focus, heterogeneity?
Response
Thank you for your helpful comments. We revised the text to make the relation between the heterogeneity and the underlying molecular mechanism clearer. (Page 14, Line 534-537: Page 16, Line 625-626)
Comments(6-2)
L663-L680. Summarizing this part, it would be as below. AF→inflammatory cell activation→ROS→RAAS activation→ Angâ…¡→AT1R→Gq→PLC cascade→MAPK→NADPH→ROS over production→MAPKMAPK→TGFβ→SMAD signaling→profibrotic process→AF vulnerability. This signaling seems looped from AF to AF. Please make it clear in the manuscript.
Response
We appreciate your good suggestion. We corrected documents to make the signal pathway clearer according to the concept of "AF begets AF." (Page 15, Line 562-563)
Comments(6-3)
L650: [202]. It showed that TBX2 tg mice had longer APD. L696: reduces APD. Does this mean shortened APD? If these are correct, how could this opposite heterogeneity, which is longer and shorter APD, cause a single consequence as AF?
Response
Thank you for your constructive question. Your advice is correct. As you commented, it is so confusing that we removed the text referring to Tbx5-mutant mice. (Page 16, Line 622)
Comments(6-4)
L710-L716. Could the opposite effects on the same molecule Cx43 by the same protein modification, phosphorylation, by PKA and PKC be mentioned as heterogeneity? It could include heterogeneities of kinase effects, spatiotemporal expression of PKA and PKC, and phosphorylated sites of Cx43 (if the site is not identical).
Response
We appreciate your comments. The different function of PKA and PKC leading to the downstream function of Cx43 depends on the phosphorylation sites. We denoted the phosphorylated site of Cx43 not to confuse the readers. (Page 14, Line 517-522)
Comments(6-5)
Related to Figure 3C, the authors could describe or hypothesize what kind of heterogenic expression/distribution of pharmacological molecular targets are assumed as well as the drugs they described.
Response
We appreciate your comments. We could describe the pharmacotherapeutic target for heterogeneity of such as profibrotic process or connexin remodeling under some pathological states. We made it clearer. (Page 21, Line 824-827)
Minor points
Comments(0)
About formatting sentences and paragraphs. Each title of the section uses italic and bold letters. Please unify the style. The space between paragraphs and sentences seems not unified in the given manuscript file. Please unify them.
Response
Thank you for your comments. We unified the font style and the spaces between paragraphs and sentences.
Comments(1-1)
L42: Is the space between “elucidated” and “These” wider than others?
Response
Thank you so much. We fixed it. (Page 2, Line 42)
Comments(1-2)
In Figure 1A, please provide what the abbreviation RAA is.
Response
Thank you for your pointing out. We added the abbreviation for RAA in the legend of Figure 1. (Page 2)
Comments(5)
L485: ERP. Please provide what the abbreviation ERP is.
Response
Thank you for your pointing out. We provided the abbreviation for ERP. (Page 17, Line 651)
Comments(6)
L689: miR. If correct, the generic name for microRNA is miRNA, although some categories of individual miRNAs are described as miR-[number].
Response
Thank you for your comment. We removed the section “Molecular regulation of inward rectifier current” as we received the comments from Reviewer 3.
Comments(7-1)
Paragraph of L734. Please insert Figures 3A, 3B and 3C into the paragraph. For example, L735 can start with “The first way (Figure 3A) is …” It makes it easier to show where each figure is explained.
Response
We inserted Figure 4A (Page 21, Line 800), 4B (Page 21, Line 812) and 4C (Page 21, Line 824) into the paragraph where each figure is explained.
Comments(7-2)
Figure 3C. There is no explanation of red dotted circles as other symbols. Also, how those can be pharmacologically heterogeneous is one of the most intriguing points in this section.
Response
Thank you for your comment. We added the documents explaining the red dotted circles in the Figure 4C and in its legend. (Page 22)
Reviewer 2 Report
Comments and Suggestions for Authors
Dear authors,
Congratulations on a simple yet very comprehensive review. The article includes a large portion of theoretical background, which can lead to a better understanding of cardiac ablation of PVs and its modifications. The interventions mentioned include novelties such as the possible impact of SGLT-2i. The article provides a comprehensive review of the heterogeneity of atrial fibrillation pathophysiology. The article focuses on including less typical localization of ectopic sources of atrial fibrillation, including the influence of the oblique ligament of Marshall or other parts of LA, including the ectopy from LAA. In contrast to previous publications, current manuscripts include heterogeneity involving the electrical activity and of anatomical interdependency, which is essential during cardiac ablation. The figures gather information regarding the division of the heterogeneities, shortly providing large portions of knowledge and showing the clinical necessity for combining the few treatment methods for atrial fibrillation. The references include 220 positions, with 1 auto citation involving the ARNI effect on atrial fibrillation in line 765—an interesting subject considering novel treatment.
After minor formatting and English correction, it should be considered for publication. The eventual changes could involve gathering the gaps in knowledge and future perspectives
Comments on the Quality of English Language
Minor editing of English language required.
Author Response
Comments
Congratulations on a simple yet very comprehensive review. The article includes a large portion of theoretical background, which can lead to a better understanding of cardiac ablation of PVs and its modifications. The interventions mentioned include novelties such as the possible impact of SGLT-2i. The article provides a comprehensive review of the heterogeneity of atrial fibrillation pathophysiology. The article focuses on including less typical localization of ectopic sources of atrial fibrillation, including the influence of the oblique ligament of Marshall or other parts of LA, including the ectopy from LAA. In contrast to previous publications, current manuscripts include heterogeneity involving the electrical activity and of anatomical interdependency, which is essential during cardiac ablation. The figures gather information regarding the division of the heterogeneities, shortly providing large portions of knowledge and showing the clinical necessity for combining the few treatment methods for atrial fibrillation. The references include 220 positions, with 1 auto citation involving the ARNI effect on atrial fibrillation in line 765—an interesting subject considering novel treatment. After minor formatting and English correction, it should be considered for publication. The eventual changes could involve gathering the gaps in knowledge and future perspectives
Response
We appreciate your nice comments. We made the manuscript corrected by the English proofreading service.
Reviewer 3 Report
Comments and Suggestions for Authors
The authors approach a complex and challenging topic.
Major comments:
.....................................
- The manuscript structure follows a Book-Chapter style more than a Review paper in a Journal. I think the authors may restructure the text; to be more concise and focus on the "Heterogeneity" concept in Atrial Fibrillation.
- Classifying heterogeneity levels is an interesting concept (right part of the first figure). However, many parts of the paper, such as Molecular, genetic, phosphorylation, etc, may not be strictly related to the topic and can be removed or at least reduced so as not to deviate or confuse the reader.
- The number and quality of figures are not satisfactory for this Review. I encourage authors to include for example more true anatomical figures (macroscopic and mesoscopic level).
- Clinical applications or targets for therapy and future directions can be improved.
Minor comments:
.....................................
- Endocardial/epicardial asynchronous activation can be also considered as EP heterogeneity at 3D level. This concept is interesting and reported by several studies from De Groot Group.
- The English language needs editing, especially in the first third of the text.
Comments on the Quality of English Language- The English language needs editing, especially in the first third of the text.
Author Response
Major comments
Comments(1)
The manuscript structure follows a Book-Chapter style more than a Review paper in a Journal. I think the authors may restructure the text; to be more concise and focus on the "Heterogeneity" concept in Atrial Fibrillation.
Response
Thank you for your constructive comment. We reconstructed the text to be concisely focusing on the “Heterogeneity” from the viewpoint of microscopic scales as well by integrating the chapter 4. “Microscopic heterogeneity” and the chapter 6. “Molecular basis predisposing to atrial fibrillation”, and by mentioning the connection between molecular mechanism and the underlying heterogeneities. (Page 13, Line 471- Page 16, Line 631)
Comments(2)
Classifying heterogeneity levels is an interesting concept (right part of the first figure). However, many parts of the paper, such as Molecular, genetic, phosphorylation, etc, may not be strictly related to the topic and can be removed or at least reduced so as not to deviate or confuse the reader.
Response
Thank you for your sharp comment. We removed the chapter “Molecular regulation of inward rectifier potassium current” and modified the text in the other molecular chapter mentioning connexin, fibrosis and genetics to make the relation to its underlying heterogeneity clearer. (Page 13, Line 504-508, Page 14, Line 535-538, Page 16, Line 624-627)
Comments(3)
The number and quality of figures are not satisfactory for this Review. I encourage authors to include for example more true anatomical figures (macroscopic and mesoscopic level).
Response
Thank you for your comment. We put anatomical images of atria depicted with computed tomography in Figure 2 (Page 3). We are afraid that we do not have enough time to get images of mesoscopic structure.
Comments(4)
Clinical applications or targets for therapy and future directions can be improved.
Response
We appreciate your thoughtful comment. We modified the text and the Figure 4 in the section 6 to make it concise based on the issues and clearer to the readers as another reviewer advised us to do it. (Page 21)
Minor comments
Comments(5)
Endocardial/epicardial asynchronous activation can be also considered as EP heterogeneity at 3D level. This concept is interesting and reported by several studies from De Groot Group.
Response
Thank you for informing us the intriguing studies from De Groot et al. We additionally discussed the transmural conduction in atria and referred to their articles. (Page 19, Line 745-749)
Comments(6)
The English language needs editing, especially in the first third of the text.
Response
We appreciate your comments. We made the manuscript edited by the English proofreading service.
Round 2
Reviewer 3 Report
Comments and Suggestions for Authors
thank you for addressing my considerations